# Low-Temperature Mineralisation of Titania-Siloxane Composite Layers

**Tomáš Svoboda [1],*, Michal Veselý [1], Radim Bartoš [1], Tomáš Homola [2] and Petr Dzik [1]**

1   Faculty of Chemistry, Brno University of Technology, Purkyňova 118, 612 00 Brno, Czech Republic;
    vesely-m@fch.vut.cz (M.V.); xcbartosr@fch.vutbr.cz (R.B.); dzik@fch.vut.cz (P.D.)
2   Department of Physical Electronics, Faculty of Science, Masaryk University, Kotlářská 267/2, 611 37 Brno,
    Czech Republic; tomas.homola@mail.muni.cz
*   Correspondence: xcsvobodato@fch.vut.cz

**Abstract:** This paper deals with low-temperature mineralisation of coatings made with titania-siloxane compositions (TSC). Methyltriethoxysilane has been adopted as the precursor for the siloxane, and during its synthesis, an oligomeric siloxane condensate with methyl moieties acting as $TiO_2$ binder has been produced. These methyl moieties, contained in TSC, provide solubility and prevent gelling, but reduce the hydrophilicity of the system, reduce the transfer of electrons and holes generated in the $TiO_2$. In order to avoid these unfavourable effects, TSC mineralisation can be achieved by nonthermal treatment, for example, by using UV-radiation or plasma treatment. Characterisation of the siloxane was performed by gel permeation chromatography (GPC), which showed the size of the siloxane chain. Thermogravimetric analysis revealed a temperature at which the siloxane mineralises to $SiO_2$. Printed layers of two types of TSC with different siloxane contents were studied by a scanning electron microscope (SEM), where a difference in the porosity of the samples was observed. TSC on fluorine-doped tin oxide (FTO) coated glass and microscopic glass were treated with non-thermal UV and plasma methods. TSC on FTO glass were tested by voltammetric measurements, which showed that the non-thermally treated layers have better properties and the amount of siloxane in the TSC has a great influence on their efficiency. Samples on microscopic glass were subjected to a photocatalytic decomposition test of the model pollutant Acid orange 7 (AO7). Non-thermally treated samples show higher photocatalytic activity than the raw sample.

**Keywords:** titanium oxide; methyltriethoxysilane; siloxane; plasma treatment; UV treatment; AO7





## 1. Introduction

The photocatalytic layers are nowadays of great interest for water [1] and air [2] purification. Titanium dioxide ($TiO_2$) layers are particularly popular because of their relatively high efficiency, non-toxicity, and affordability. $TiO_2$ have been widely applied for contaminant remediation and microorganism destruction [3,4].

$TiO_2$ is often used as a coating on hard and durable materials such as glass. Recently, there has been an interest in depositing it on flexible substrates such as polyurethanes, polyesters, polyvinyl chlorides, and others, for photovoltaic, textile, and paper industries, and others. The main problem is the stabilisation of $TiO_2$ on the substrate in order not to release it from the photocatalytic layer into the environment [4–6]. Generally, thermal methods of hundreds of degrees Celsius [7] are used to increase the adhesion and photocatalytic activity. Another problem lies in the $TiO_2$ photocatalytic activity that disturbs the organic materials onto which $TiO_2$ is deposited. The main directions of the research are to protect the substrates from UV and photocatalytic degradation, to ensure sufficient penetration of the pollutants to the photocatalyst, to provide flexibility, and to develop the process of preparing such a photocatalytic system that could be used on an industrial scale [6,8].

One way to protect the substrate from degradation by thermal treatments and photo-catalytic processes is to use a suitable binder as a matrix. The matrix serves as a mechanical support for $TiO_2$ and as a protective layer for the substrate. However, the bonding of the binder itself may not be a sufficient condition for a functional photocatalytic layer. Other treatments are needed that may also be thermal treatments, but it would be possible to use the photocatalytic processes themselves in the layer using UV-irradiation or short-term exposure to plasma for low-temperature treatment.

Many areas extensively use mesoporous oxide thin films as functional and structural materials. They include protective and low-dielectric constant layers, selective gas permeation membranes, wettability layers, and gas sensors [9–11].

In this work we developed and investigated polysiloxane as the $TiO_2$ anchor matrix and we obtain the titania-siloxane composition (TSC), which can be used for deposition on flexible substrates without the need for thermal treatments which damage the substrate. Siloxane (oligomer or polymer) can be obtained from organosilicon precursors, which can be further doped with$TiO_2$ particles. Siloxane serves as a matrix for anchoring $TiO_2$, and, at the same time, it can retain porosity to successfully adsorb the pollutants to the surface of the photocatalyst grains deposited deeper in the layer [6,12]. The siloxane used in this study contains a certain proportion of methyl moieties that provide solubility and prevent gelling. On the other hand, the methyl moieties reduce the transfer of electrons generated by the $TiO_2$ and significantly deteriorates the photocatalytic activity of $TiO_2$. There is also a certain proportion of hydroxyl groups in siloxanes. The hydroxyl group, as a residual group, induces moisture adsorption through hydrogen bonding when exposed to moisture [13]. Moisture ($H_2O$) is an important part of the photocatalytic processes taking place in titanium dioxide. Moisterure-binding hydroxyl groups improve the wettability of the surface and the photocatalytic activity itself and is a better supply of pollutants in the aqueous medium [14]. In order to remove the organic matter from the siloxane and improve the transfer of the electrons, siloxane must be completely mineralised towards amorphous silica surface. This step is traditionally performed by thermal annealing at hundreds of °C. On the other hand, this approach is not compatible if flexible and thermally-sensitive substrates such as polyethylene terephthalate (PET) and polyethylene naphthalate (PEN) are used. Low-cost substrates are favorable for future generation manufacture of emerging technologies, including flexible and printed electronics. Therefore, it is important to investigate novel low-temperature methods compatible with rapid and low-temperature post-treatment of siloxanes as replacements for traditional thermal annealing that is not compatible with flexible electronics.

In this work we also investigated two non-thermal methods for post-processing of titania-siloxane composition (TSC) layers: UV-irradiation and open-air plasma. UV-irradiation was already successfully tested for the fabrication of amorphous $TiO_2$ thin films. UV-irradiation at room temperature leads to a higher conduction band minimum level of the film and a smaller amount of hydroxyl group at the film surface, compared to the thermal-assisted (100–250 °C) UV-annealing or the thermal-only annealing (500 °C). [15]. Plasma treatment can be used for calcination and removal of organic residues from sol–gel and generation of mesoporous films [16–20]. The plasma technique is more attractive because it has many advantages, such as low processing temperature, short processing time and inexpensive equipment [9].

We studied the properties of TSC in synergy with UV-irradiation and plasma treatment as the techniques for non-thermal curing of photocatalytic layers to improve photocatalytic activity. The main parameter for the mineralisation of siloxane was the study of the decrease of methyl groups by the Fourier-transform infrared spectroscopy (FT-IR) method. The photocatalytic activity was monitored by voltammetric measurements and photocatalytic degradation of AO7.

## 2. Results and Discussion

### 2.1. Viscosity of Siloxane Solutions

Figure 1 shows viscosity of siloxane solutions tested with various concentration of methanol ranging from 10% to 50%. All solutions tested show linear dependence of dynamic viscosity with shear rate. The sample with 50% of ethanol showed a slight deviation from Newtonian behaviour (at the lowest shear rate there is a sign of viscosity increase). The viscosity of the solutions increases considerably as the concentration increases, which is important, especially for the future printing of compositions containing siloxane.

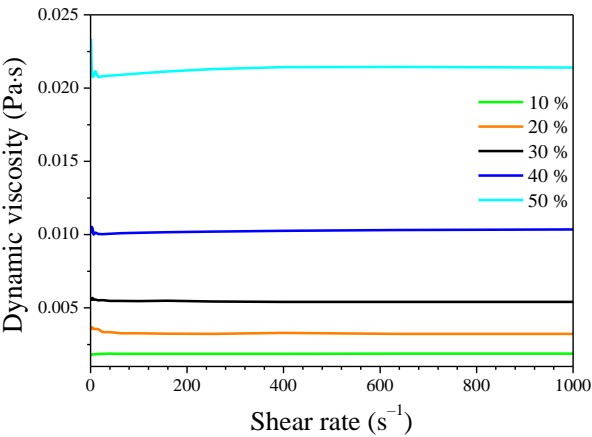

**Figure 1.** Viscosity of siloxane solutions in absolute ethanol (concentrations 10, 20, 30, 40, 50%).

### 2.2. Surface Tension of Siloxane Solutions

We also investigated the effect of applying concentrations of ethanol on the surface tension of siloxane solutions. The pure ethanol has a surface tension of 21.90 $mN \cdot m^{-1}$ at 25 °C [21]. The results in Table 1 show that there was no significant change in the surface tension of solutions with various concentration of ethanol. However, with the decreasing concentration of pure ethanol, a slight increase in surface tension occurs when the concentration of siloxane in volume, and hence on the surface of the liquid, increases. We see a certain similarity with ethanol, considering the possible structure of siloxane (Equations (1) and (2)) where $R-Si-(OH)_3$ is present in the solution, and because in our case R means methyl, the hydrocarbon residue will not have the weight to significantly affect the surface tension. It can therefore be assumed that the surface tension of the resulting siloxane-blended compositions will not be significantly affected by the amount of siloxane.

**Table 1.** Surface tension of siloxane solutions (concentrations 10, 20, 30, 40, 50%).

| Concentration (%) | 10 | 20 | 30 | 40 | 50 |
|---|---|---|---|---|---|
| Surface tension ($mN \cdot m^{-1}$) | 22.58 | 22.96 | 23.52 | 24.08 | 24.75 |

### 2.3. Gel Permeation Chromatography

Gel permeation chromatography (GPC) results are shown in Table 2. The polydispersity is close to 1, which means that the particles in solution have similar dimensions, expressed by the radius of gyration and the weight average molar mass.

**Table 2.** The results of GPC. The polydispersity is close to 1, which means that the particles in solution have similar sizes.

| Dilution of 10% Solution | Injection Volume (μL) | $M_W$ (kDa) | Polydispersity ($M_W/M_N$) | Radius of Gyration (nm) |
|---|---|---|---|---|
| 1:1 | 100 | 1.497 | 1.019 | 12.8 |
| 1:1 | 100 | 1.534 | 1.036 | 11.9 |
| 1:1 | 100 | 1.475 | 1.033 | 12.0 |
| 1:1 | 100 | 1.506 | 1.046 | 12.5 |
| | Diameter $M_W$ (kDa) | 1.52 | 1.032 | 13 |
| | Selective standard deviation | 0.04 | 0.009 | 2 |

### 2.4. Specific Surface Area (SSA) and SEM of Siloxane/TiO₂

Two siloxane/TiO$_2$ compositions of ratios 1:1 and 1:3 were created and compared. SSA was measured by by nitrogen adsorption using the BET isotherm (Figure 2). The two different values of siloxane/TiO$_2$ ratios have a direct impact of the layer textural properties as is evident from Figure 3. The more titania rich formulation exhibited a fluffy texture with many accessible voids while the more binder rich formulation is apparently denser and more compact. SSA for siloxane/TiO$_2$ compositions of ratios 1:1 and 1:3 was calculated as 12.3 m$^2$/g and 34.9 m$^2$/g, respectively. The question of phase composition of both coatings was addressed separately in our previous communication [22] and we found out that binder mineralisation process has no impact on the crystallinity of titania.

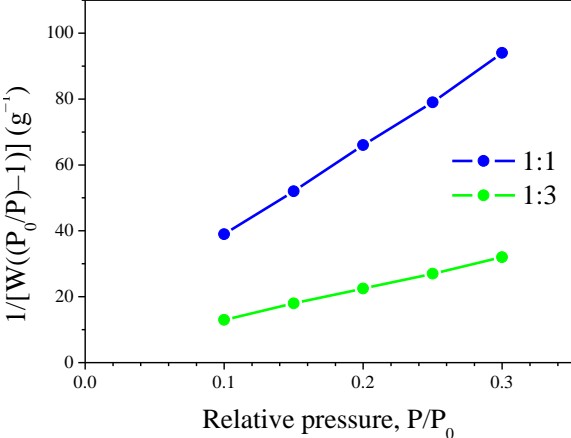

**Figure 2.** Multi-point BET plot for siloxane/TiO$_2$ with various ratio 1:1 and 1:3.

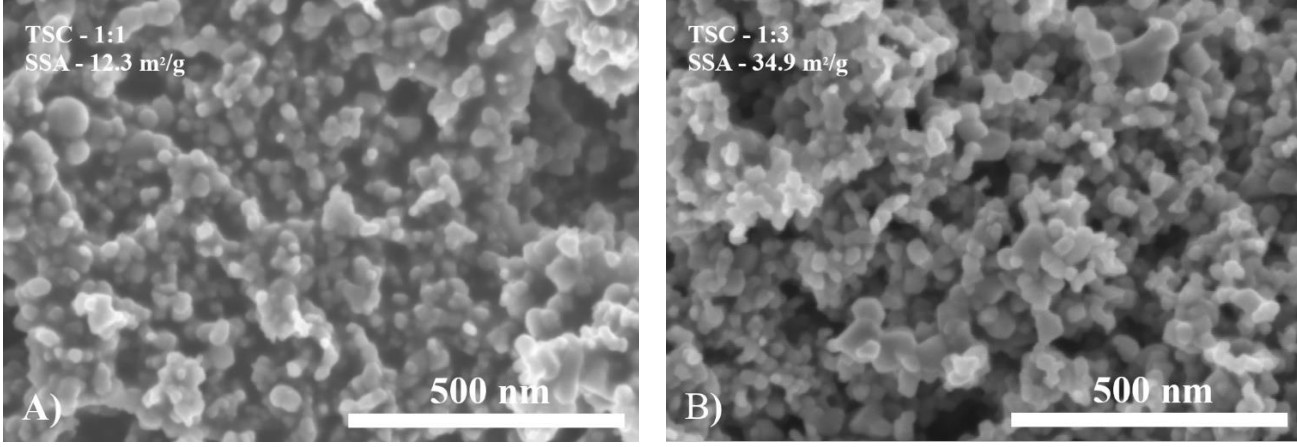

**Figure 3.** SEM images. (**A**) Shows a less porous layer with a siloxane/TiO$_2$ ratio of 1:1, and (**B**) shows a much more porous layer of a 1:3 ratio.

### 2.5. Thermal Treatment of TSC and TGA, DTG

The coatings were further sintered in furnace oven at temperature 450 °C for 30 min to mineralise the binder, i.e., remove the organic methyl moieties and support the transport of electrons from $TiO_2$. FT-IR results showed in Figure 4 revealed that siloxane was not mineralised completely. Peaks located at $2995-2950$ cm$^{-1}$ and $2895-2840$ cm$^{-1}$ corresponding to asymmetrical and symmetrical stretching of $-CH_3$ and indicating the presence of methyl groups in polysiloxane binder.

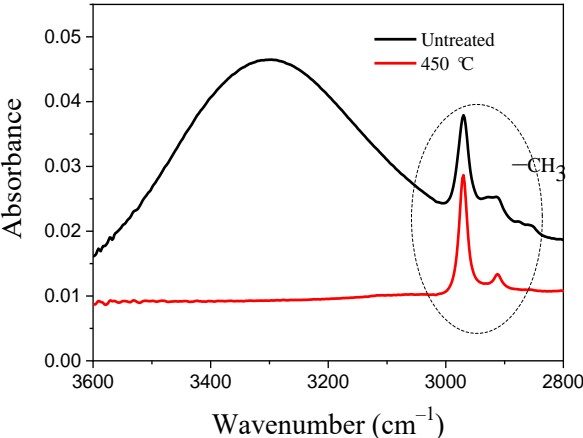

**Figure 4.** FT-IR measurement of TSC on soda-lime glass. The processing temperature of 450 °C was not sufficient to mineralise the siloxane because the methyl groups are still present after thermal treatment.

The coatings were further analysed by DSC-TGA and heated to 1300 °C. During the thermal process, there were several changes as the temperature rose (Figure 5). The first range was from 20 °C to 477 °C. There was probably a loss of sorbed air moisture or residual solvent. Weight decreases steeply. The second major change occurs at 478 °C, when there was a further rapid drop in weight. Siloxane should be calcined at this temperature and thus degrade the primarily methyl groups bonded to silicon oxide. Another weight change was significantly slowed down from a temperature of 600 °C until it stabilised and no changes occurred. It is clear from the measurements that, for complete calcination, it is necessary to achieve temperatures higher than 478 °C, and therefore, the thermal method of mineralisation is only suitable for resistant materials.

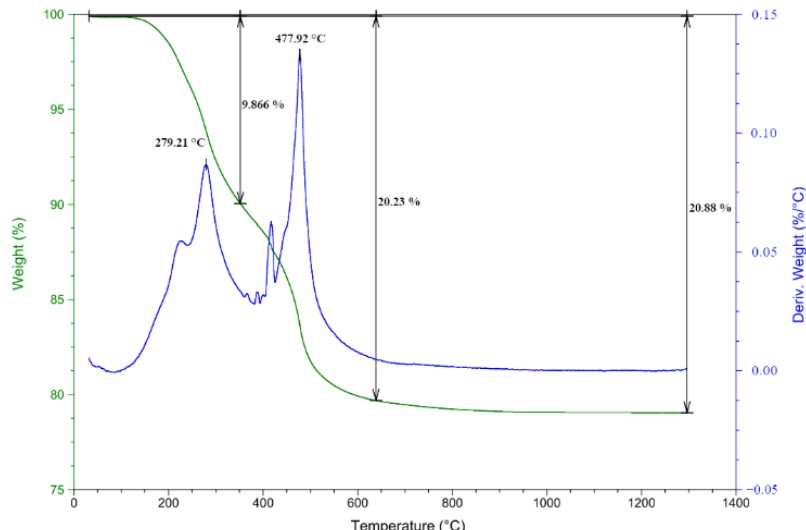

**Figure 5.** The results of the TGA, DTG. The mineralisation and removal of the methyl groups from the siloxane should occur at 478 °C.

### 2.6. FT-IR of Nonthermal Curing TSC

UV-irradiation and non-thermal atmospheric-pressure plasma treatment were further investigated in order to replace time-consuming thermal sintering, which is problematic if polysiloxane/$TiO_2$ is deposited on thermally-sensitive materials. Both siloxane/$TiO_2$ compositions of ratios 1:1 and 1:3 were investigated by FT-IR before and after UV-irradiation and plasma treatment.

Figure 6A,B shows FT-IR spectra of siloxane/$TiO_2$ layers prepared by both compositions of ratios 1:1 and 1:3, cured by UV-irradiation for 0–210 min. The decrease of peaks related to asymmetrical and symmetrical stretching of $-CH_3$ (2995−2950 $cm^{-1}$ and 2895−2840 $cm^{-1}$) indicate the removal of $-CH_3$ groups from siloxane surface and apparently the transformation of siloxane towards amorphous $SiO_2$. For both coatings (1:1 and 1:3 ratio), UV-irradiation for 150 min was enough to remove all detectable $-CH_3$ groups. This decrease can be further explained by photocatalytical reaction between anatase $TiO_2$ with bandgap 3.2 eV [23] that can provide additional catalytical reaction on polysiloxane surface and enhance the degradation of methyl groups.

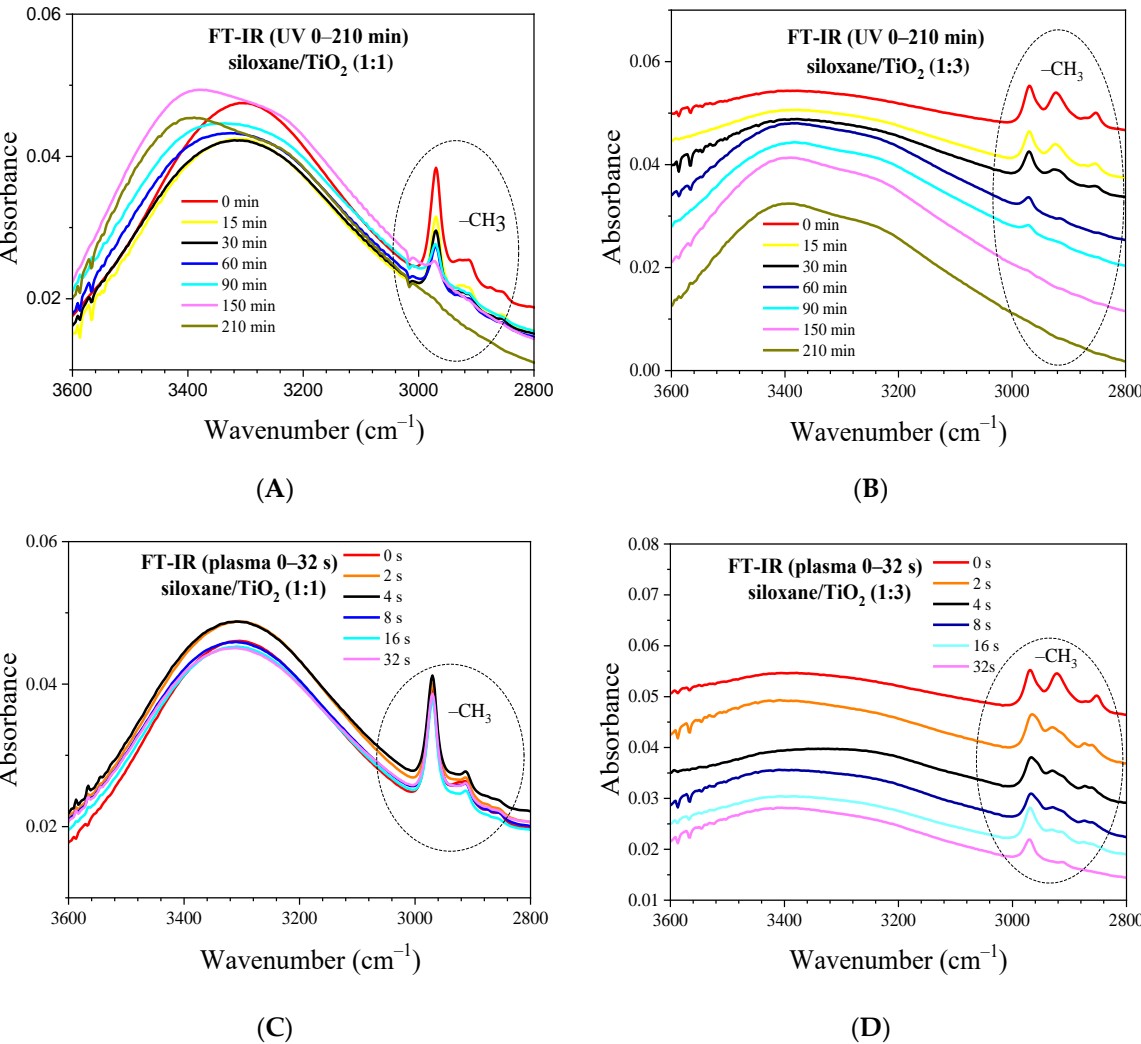

**Figure 6.** FT-IR measurements of TSC layers. (**A**,**B**) are images for samples treated with UV-irradiation; (**C**,**D**) are images for plasma-treated samples. UV-irradiation has better layer penetration and is therefore effective for layers with a higher concentration of siloxane.

Figure 6C,D shows FT-IR spectra of siloxane/$TiO_2$ layers prepared by both compositions of ratios 1:1 and 1:3, cured by plasma for 0–32 s. In contrast with the results presented for UV-irradiated layers, the plasma treated layers showed significantly lower efficiency

of methyl groups removal. The most profound difference was found for the film of lower porosity (1:1) and higher efficiency was found for the layer of higher porosity. This discrepancy is clearly related to the difference in mechanism of UV-irradiation and plasma treatment. Whereas the dominant energy-transfer mechanism in UV-irradiation is the transport of photons of certain wavelength into the porous film, the plasma treatment works differently. Apart from the UV-irradiation, the plasma generated at atmospheric pressure in ambient air contains various energetic species that can carry energy towards siloxane surface: High-temperature electrons, low-temperature ions, excited species and metastables. These species, however, recombine and extinct in contact with materials surface and thus, the porosity of material plays an important role in the limitation of such method. Since plasma treatment was less efficient for methyl removal at less porous material, it can be concluded that plasma cannot efficiently penetrate into the coating bulk and mineralise it. On the other hand, the coating with high porosity showed better efficiency of methyl removal, and therefore, it is important to find an optimal combination of layer porosity if atmospheric pressure plasma is used for mineralisation of coatings. Although not efficient as UV-irradiation, a clear benefit of plasma treatment is in significantly faster treatment times, in order of ten seconds, which allow to use this method on fast roll-to-roll production lines and employ the coatings in flexible and printed electronics concept.

### 2.7. Voltammetric Measurements of Nonthermal Curing TSC

Voltammetric measurements can be taken for the photocatalytic activity test. If the treated layer is exposed to UV-irradiation, it generates an electron/hole pair. By applying the external voltage, the electrons can be abstracted from the exciton and the photocurrent is detected in the external circuit. The growth of the photocurrent is related to higher photocatalytic activity [24].

The siloxane/$TiO_2$ films of ratios 1:1 and 1:3 were exposed to UV-irradiation for 0–120 min and plasma treatment for 0–32 s. Figure 7 shows the voltametric measurement of photocurrent generated by the coating upon exposure to UV. The UV-irradiation pretreatment for 0–120 min led to a gradual increase of the photocurrent for both coatings (1:1, 1:3) investigated. Higher values of photocurrent were measured for coating with a higher concentration of $TiO_2$ in siloxane/$TiO_2$ (ratio 1:3). This is apparently thanks to higher concentration of photo-catalytically active $TiO_2$ that can not only improve the mineralisation reaction, but also supply more electrons upon UV exposure. Furthermore, the porosity of the coating with siloxane/$TiO_2$ ratio 1:3 is higher, so the higher photocurrent is most likely an interplay between these factors.

Figure 7C,D shows voltammetric measurements of siloxane/$TiO_2$ films of ratios 1:1 and 1:3 were exposed to air plasma for 0–32 s. The coating with small porosity yields very small photocurrents, and it seems that plasma treatment had a negative effect on the maximal photocurrent values. On the other hand, the plasma treatment of the coating with higher porosity (1:3) resulted in a gradual increase in photocurrent with plasma-exposure time. The better efficiency in photocurrent generation in the coating of higher porosity is related to higher mineralisation efficiency, as shown in FT-IR results presented in Figure 6D.

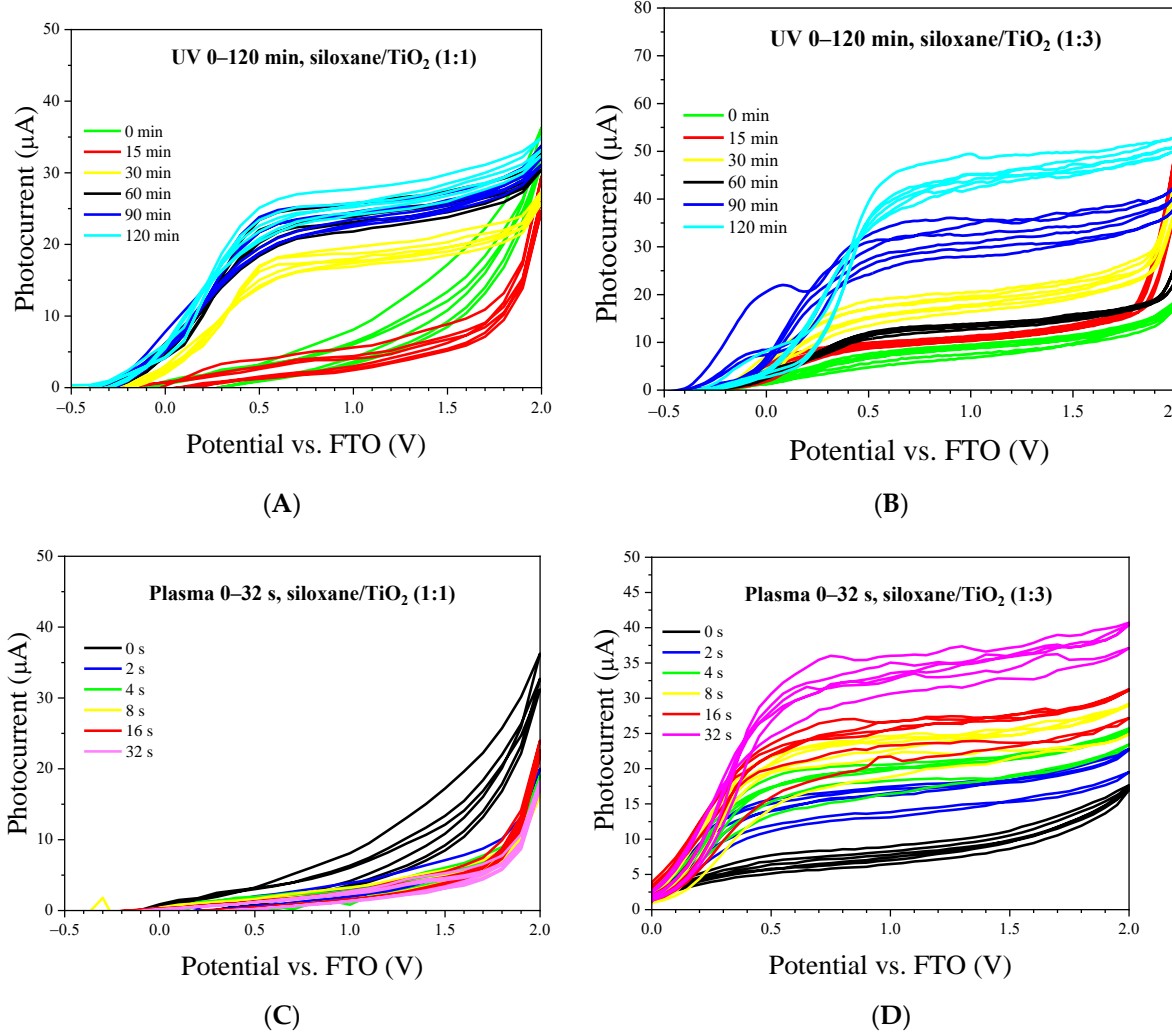

**Figure 7.** Voltammetric characterisation of TSC printed on FTO-coated glass. (**A**,**B**) are images for samples treated with UV-irradiation; (**C**,**D**) are images for plasma-treated samples. Plasma can not penetrate the layer sufficiently if it contains too much siloxane. In the case of (**C**), the photocurrent is very low to zero.

### 2.8. Photocatalytic Degradation of Acid Orange 7

Acid Orange 7 (AO7) serves as a model pollutant for the photodegradation test. AO7 turns the aqueous solution to orange and it is a resistant substance that slow degrades. Figure 8 shows the decrease in absorbance signal reflecting the concentration of AO7 upon irradiation of UVA for 1 h. UVA was chosen because radiation with lower wavelengths is less advantageous in terms of energy efficiency. We compared three samples with ratio 1:3: untreated siloxane/$TiO_2$, UV-irradiated siloxane/$TiO_2$ for 210 min and plasma-treated siloxane/$TiO_2$ for 32 s. The slowest degradation of AO7 was observed for untreated siloxane/$TiO_2$, whereas the fastest degradation was observed for UV-irradiated siloxane/$TiO_2$ for 210 min. The plasma-treated siloxane/$TiO_2$ showed significant improvement in AO7 degradation in comparison to the untreated sample, although the efficiency of siloxane/$TiO_2$ irradiated by UV was better. It should be noted that UVA involved in AO7 degradation test may further contribute to mineralisation of TSC [25].

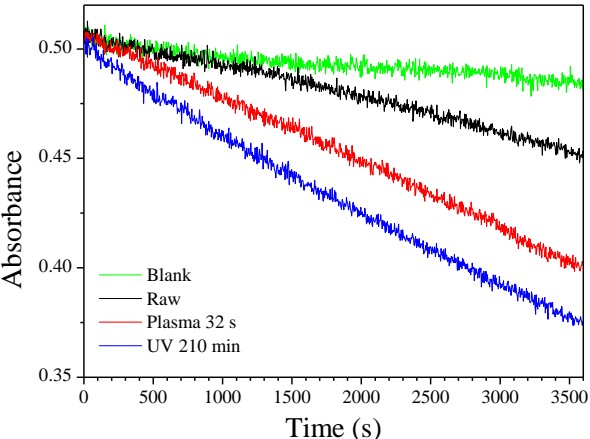

**Figure 8.** Degradation of 2 mg/L aqueous solution of AO7 with immobilized TSC layers treated by plasma and UV-irradiation.

## 3. Materials and Methods

### 3.1. Synthesis of Siloxane

The starting substance for the synthesis of siloxane was methyltriethoxysilane (MTEOS) (Alfa Aesar, 98%, Haverhill, MA, USA). MTEOS was hydrolysed with acidic water. Ethanol was formed during the hydrolysis and subsequently distilled. Siloxane was extracted with diethyl ether (Penta, 99.7%), and it was dissolved in absolute ethanol (Penta, 99.8%, Prague, Czech Republic) after evaporation of the extracting agent. The siloxane solution in ethanol was stored at a temperature below 0 °C [6]. The expected chemical reaction can be summarised by the following equation [26]:

$$R - Si(OC_2H_5)_3 + 3H_2O \rightarrow R - Si(OH)_3 + 3C_2H_5OH \tag{1}$$

$$2R - Si(OH)_3 \leftrightarrow R - Si(OH)_2 - O - Si(OH)_2 - R + H_2O \tag{2}$$

### 3.2. Preparation of TSC

Two series of samples were prepared for experiments, as shown in Table 3. $TiO_2$ (P25, Sigma Aldrich, 99.7%, St. Louis, MO, USA, particle size $\leq$ 25 nm, SSA 45–55 m$^2$/g [27]) was dispersed in dowanol and a siloxane 20% solution was added. The resulting suspension was further diluted with hexanol. The TSC were printed with a Dimatix (DMP-2800) material printer on soda-lime glass and FTO-coated glass.

**Table 3.** Compositions designed for material printing.

| Composition | Siloxane (20% in Ethanol) | TiO$_2$ (20% P25 in Dowanol) | Hexanol |
|---|---|---|---|
| E38-9AD (1:1) | 4 mL | 4 mL | 20 mL |
| E38-10AD (1:3) | 2 mL | 6 mL | 20 mL |

### 3.3. Mineralisation of the Printed TSC on the Substrate

The siloxane/$TiO_2$ coatings were mineralised by two methods: UV-irradiation and plasma treatment. UV-irradiation was generated by a Sylvania UV lamp Figure 9 (mercury, 125 W, Budapest, Hungary). The radiation intensity was set and held at 9 mW·cm$^{-2}$. Samples were placed under 5 mm of distilled water and irradiated for 0, 15, 30, 60, 90, 150, and 210 min.

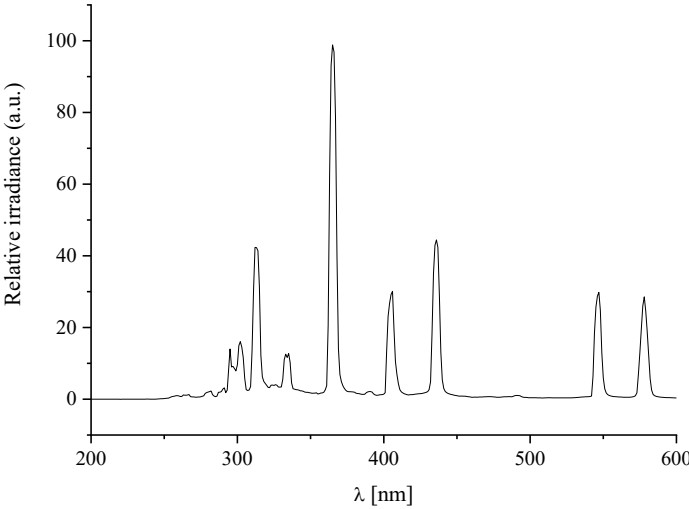

**Figure 9.** Spectrum of the Sylvania UV lamp (mercury, 125 W).

The plasma treatment was performed by RPS400 (Roplass s.r.o., Brno, Czech Republic) equipped by dielectric barrier discharge (DBD) with a coplanar arrangement of the electrode system: A diffuse coplanar surface barrier discharge (DCSBD) plasma unit. The DCSBD is capable of generating a thin surface plasma of very high-power density up to 100 W·cm$^{-3}$ at very low temperature ~70 °C in open air environment at atmospheric pressure. The details of the DCSBD plasma can be found in Homola et al. [28,29]. The plasma exposure times of siloxane surfaces in this study were 0, 2, 4, 8, 16 and 32 s.

*3.4. Characterisation of Siloxane*

The properties of siloxane solutions in ethanol were studied by means of various methods. Since the ink-jet printing is compatible only to a certain range of viscosity (10–30 cP) and other rheologic parameters, it is important to determine the viscosity, surface tension, and size of agglomerates, in order to verify if the compositions have the desired properties and do not suffer complications during printing.

The concentration range of siloxane (10, 20, 30, 40, 50%) was used for viscosity measurements carried out on a rotating rheometer AR-G2 (TA Instruments) with a roller sensor in a cylinder at 25 °C. The surface tension was measured by tensiometer KSV Sigma 701 using the *du Noüy ring method* with a platinum ring.

The SEC-MALS-dVI-dRI system was used for GPC measurements. Chromatographic part (Agilent Technologies, Santa Clara, CA, USA): consisting of isocratic pump, degasser, autosampler, column and UV-VIS detector. Detectors (Wyatt Technology, Dernbach, Germany): DAWN HELEOS II: multi-angle light scattering (MALS), VISCOSTAR II: differential viscometer (dVI), OPTILAB T-REX: differential refractometer (dRI). The 10% siloxane solution was further diluted with absolute ethanol 1:1.

The thermal gravimetric analysis was performed with Instrument SDT Q600. The siloxane solution was dried to 100 °C and the solid was exposed to a temperature of up to 1300 °C in the air with a step 10 °C/min. Thermal analysis was also performed to give us insight into changes during the thermal mineralisation of siloxane.

*3.5. Characterisation of TSC and Printed Layers on Glass*

Brunauer–Emmett–Teller (BET) method was employed to determine the specific surface area (SSA) of TSC. BET was performed on Autosorb iQ. Prior to BET analysis, the samples were dried at 100 °C.

FT-IR Nicolet iS5 was used in order to measure the decrease the methyl groups after non-thermal curing by UV-irradiation and plasma treatment. The TSC samples on soda-lime glass were measured on FT-IR using the Omnic program. The spectral region characteristic of the 2950 cm$^{-1}$ and 2895–2840 cm$^{-1}$ methyl groups was observed [30].

The photo-electrochemical characterization of TSC on the FTO-coated glass was performed by linear sweep voltammetry at room temperature using a two-electrode setup with the 1 cm$^2$ titania patches. The printed FTO slide was scratched with a diamond knife and thus two isolated FTO strips were created. One strip with the printed titania patch, served as the working electrode and the opposite naked FTO strip as the counter electrode. This setup was fitted into a custom build quartz cuvette. The cuvette was filled with 0.1 M perchloric acid (conductivity 36 mS·cm$^{-1}$) and fitted onto an optical bench equipped with a fluorescent UV-A lamp emitting a broad peak centered at 365 nm (Sylvania Lynx-L 11 W). A magnetic stirrer was placed beneath the cuvette and a magnetic flea inside the cuvette provided efficient electrolyte mixing. The lamp emission was monitored by Gigahertz Optic X97 Irradiance Meter with a UV-3701 probe and the irradiance was set to 2 mW·cm$^{-2}$ by adjusting the lamp-to-cuvette distance. Measurements of generated photocurrents were performer with an electrometer build on the basis of National Instruments Labview platform and supplying a linear voltage gradient of 10 mV/s from −0.5 to 2 V and measuring the generated currents down to submicroampere range [24].

Photocatalytic degradation test of TSC on glass with 12 cm$^2$ E38-10AD patches was performed by degradation recrystallization AO7 (Centre for Organic Chemistry) solution in water. The quartz cuvette was filled with 2 mg/L aqueous solution of AO7 and fitted onto an optical bench equipped with a fluorescent UV-A lamp emitting a broad peak centered at 365 nm (Sylvania Lynx-L 11 W). The absorbance of the resulting solution at a 4 cm optical path was 0.5 at a wavelength of 480–490 nm. A magnetic stirrer was placed beneath the cuvette and a magnetic flea inside the cuvette provided efficient solution mixing. The lamp emission was monitored by Gigahertz Optic X97 Irradiance Meter with a UV-3701 probe and the irradiance was set to 2 mWcm$^{-2}$ Measurements of decrease in absorbance AO7 were performer with a spectrometer Red Tide USB650 (Ocean Optics, Prague, Czech Republic) and an Ocean View program.

The TSC samples printed on FTO-coated glass were also examined by the TESCAN Mira3 XMH scanning electron microscope. Coatings were printed on FTO, in order to prevent charging and deposition of conductive layer onto TSC.

## 4. Conclusions

This contribution presents the non-thermal curing of titania siloxane coatings by two atmospheric methods. Titania siloxane coatings are promising alternative for other photo-catalytically active layers on substrates that cannot withstand processing at temperatures >150 °C, e.g., thermally sensitive polymers and paper. This work demonstrates and compares slow and efficient UV-irradiation, and extremely rapid plasma treatment, as methods for low-temperature sintering of titania siloxane photo-catalytically active coatings. It was also investigated that the main mechanism of the non-thermal curing is the removal of the −CH$_3$ methyl groups from the siloxane chain while the bulk structure of the titania content remained unaffected. Although, UV-irradiation showed more efficient and led to the removal of most of the detectable methyls from the siloxane surface, the plasma treatment resulted in much faster processing times in the order of tens of seconds. This, on the other hand, is promising for the implementation of the method into fast roll-to-roll production line. Since both methods, UV and plasma, are scalable to several meters in width, the employment of any of them or even the combination of UV and plasma treatment could be of interest development of new large scale thin-film photocatalysts.

**Author Contributions:** Conceptualization, T.S. and P.D.; methodology, P.D., T.S.; investigation, T.S.; supervision, P.D. and M.V.; writing—original draft preparation, T.S.; software, T.S.; writing—review and editing, P.D., T.H. and T.S.; data curation, T.S. and R.B.; resources, P.D., M.V. and T.H.; validation, P.D., T.H., R.B. and T.S.; project administration, T.H., M.V. and P.D. All authors have read and agreed to the published version of the manuscript.

**Funding:** This work has been supported by Czech Science Foundation research project 19-14770Y.

**Institutional Review Board Statement:** Not applicable.

**Informed Consent Statement:** Not applicable.

**Data Availability Statement:** The data presented in this study are available in the article.

**Conflicts of Interest:** The authors declare that they have no competing interests.

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
