# Peer review of "Low-Temperature Mineralisation of Titania-Siloxane Composite Layers"

_catalysts, doi:10.3390/catal11010050_

Round 1

Reviewer 1 Report

Authors present an interesting manuscript on new approaches to anchor TiO2 in low temperature conditions. Nonetheless, prior to its publication in Catalysts, some issues should be addressed. 

  1. Double check the Figures linked references, this message appears in the text "Error! Reference source not found."
  2. Check the text for some minor grammar errors and lots of misspeling mistakes such as: Line 95, it should be change, not hange. Line 107: which instead of hich. Line 114: ratio instead of atio (and many others). Please double check all the manuscript to avoid this minor mistakes.
  3. In the first part of the results and discussion section, you explain the rheological behaviour of the solutions, but the reader has no clue on which kind of solution is employed. Please make a brief introduction or include more information on the caption of Figure 2.
  4. All abbreviations should be introduced and section titles should include complete terms, rather than abbreviations. Please check section 2.3. What does GPC stand for?
  5. Materials and methods: you should specify the maximum wavelenght emission of the lamp. Is it UV-A, UV-B, UV-C?
  6. For the siloxane mineralization in the UV method,  you subjected the samples to UV-radiation for up to 3.5 h, whereas the lonngest plasma treatment lasted 32 s. What happens if you apply plasma for 2 or 5 min? It would still be a significantly lower time than that required for UV-treatment. 
  7. Caption of Figure 3 includes discussion that should go in the text rather than in the caption
  8. Materials and methods should include information on the voltammetric measurements. Used equipment, electrodes employed, etc. When representing the results, usually axis X represents potential against the reference electrode (AgCl,  for instance) please reformulate. 
  9. The introduction of the article is focused on the use of these materials for pollution depletion. Some test on the catalytic activity of the coated glass against a model contaminant would greatly enhance the quality of the paper. 

Reviewer 2 Report

The submission titled " Low-temperature mineralization of titania-siloxane 2 composite layers" introduced the titania-siloxane composition system. They conducted some analyses to characterize the samples. However, the quality of the presentation was not enough to be published. For example, line 85 on page 2 of 11, line 94 on page 3 of 11 shows "Error! Reference source not found." Moreover, the quality of SEM images (Figure 4) was not good and the scale bar must be included in the SEM images. The figure captions in Figure 6 must be enlarged. It should be re-drawn from the raw data, instead of using the image by "print screen". In addition, the peaks in FTIR should be identified the functional groups with the reference. Since the TiO2-flexible composites are popular, the overall novelty and significance of this manuscript seem not to be sufficient to be published. The authors must emphasize the novelty of their study and modify the quality of the presentation. Therefore, I won’t suggest this manuscript for publishing in the present form.

Round 2

Reviewer 2 Report

The article has been improved to a satisfactory level, therefore I can recommend publication in the present form.

Author Response

Thank you.
